# Nomadic-colonial life strategies enable paradoxical survival and growth despite habitat destruction

Zhi Xuan Tan[1], Kang Hao Cheong[2]*

[1]Yale University, New Haven, United States; [2]Engineering Cluster, Singapore Institute of Technology, Singapore, Singapore

**Abstract** Organisms often exhibit behavioral or phenotypic diversity to improve population fitness in the face of environmental variability. When each behavior or phenotype is individually maladaptive, alternating between these losing strategies can counter-intuitively result in population persistence–an outcome similar to the Parrondo's paradox. Instead of the capital or history dependence that characterize traditional Parrondo games, most ecological models which exhibit such paradoxical behavior depend on the presence of exogenous environmental variation. Here we present a population model that exhibits Parrondo's paradox through capital and history-dependent dynamics. Two sub-populations comprise our model: nomads, who live independently without competition or cooperation, and colonists, who engage in competition, cooperation, and long-term habitat destruction. Nomads and colonists may alternate behaviors in response to changes in the colonial habitat. Even when nomadism and colonialism individually lead to extinction, switching between these strategies at the appropriate moments can paradoxically enable both population persistence and long-term growth.

*For correspondence:
kanghao.cheong@singaporetech.
edu.sg

Competing interests: The authors declare that no competing interests exist.

## Introduction

Behavioral adaptation and phenotypic diversity are evolutionary meta-strategies that can improve a population's fitness in the presence of environmental variability. When behaviors or phenotypes are sufficiently distinct, a population can be understood as consisting of multiple sub-populations, each following its own strategy. Counter-intuitively, even when each sub-population follows a losing strategy that will cause it to go extinct in the long-run, alternating or reallocating organisms between these losing strategies under certain conditions can result in meta-population persistence, and hence, an overall strategy that wins (*Williams and Hastings, 2011*). Some examples include random phase variation (RPV) in bacteria across multiple losing phenotypes (*Wolf et al., 2005*; *Kussell and Leibler, 2005*; *Acar et al., 2008*), as well as the persistence of populations that migrate among sink habitats only (*Jansen and Yoshimura, 1998*; *Roy et al., 2005*; *Benaim and Schreiber, 2009*).

These counter-intuitive phenomena are reminiscent of Parrondo's paradox, which states that there are losing games of chance which can be combined to produce a winning strategy (*Harmer and Abbott, 1999*). The existence of a winning combination relies on the fact that at least one of the losing Parrondo games exhibits either capital-dependence (dependence upon the current amount of capital, an ecological analog of which is population size) or history-dependence (dependence upon the past history of wins or losses, or in an ecological context, growth and decline) (*Parrondo et al., 2000*; *Harmer and Abbott, 2002*). There have been many studies exploring the paradox (*Cheong and Soo, 2013*; *Soo and Cheong, 2013*, *Soo and Cheong, 2014*; *Abbott, 2010*; *Flitney and Abbott, 2003*; *Harmer et al., 2001*), including a multi-agent Parrondo's model based on complex networks (*Ye et al., 2016*) and also implications to evolutionary biology (*Cheong et al.,*

**eLife digest** Many organisms, from slime molds to jellyfish, alternate between life as free-moving "nomadic" individuals and communal life in a more stationary colony. So what evolutionary reasons lie behind such stark behavioral diversity in a single species? What benefits are obtained by switching from one behavior to another?

Tan and Cheong have now developed a mathematical model that suggests an intriguing possibility: under conditions that would cause the extinction of both nomadic individuals and colonies, switching between these life strategies can enable populations to survive and grow – a counter-intuitive phenomenon called Parrondo's paradox.

Parrondo's paradox says that it is possible to follow two losing strategies in a specific order such that success is ultimately achieved. For example, slot machines are designed to ensure that players lose in the long run. What the paradox says is that two slot machines can be configured in such a way that playing either slot machine will lead to financial disaster, but switching between them will leave the player richer in the long run.

Most studies of similar phenomena suggest that switching between two 'losing' lifestyle strategies can only improve the chances of survival if the environment keeps changing in unpredictable ways. However, Tan and Cheong's model shows that this unpredictability is an unnecessary condition – paradoxes also occur when organisms form colonies that predictably destroy their habitat.

The basic mechanism for survival is elegant. The organism periodically exploits its habitat as part of a colony, then switches to a nomadic lifestyle to allow the environment to regenerate. Through mathematical analysis and simulations, Tan and Cheong confirm that this strategy is viable as long as two conditions hold: that colonies grow sufficiently quickly when environmental resources are abundant; and that colonists switch to a nomadic lifestyle before allowing the resource levels to dip dangerously low.

The results produced by Tan and Cheong's model help to explain how behavior-switching organisms can survive and thrive, even in harsh conditions. Further work needs to be done to adapt this general model to specific organisms and to investigate the possible evolutionary origins of behavior-switching lifestyles.

*2016*; *Reed, 2007*; *Wolf et al., 2005*; *Williams and Hastings, 2011*). However, many biological studies which have drawn a connection to Parrondo games do not necessarily utilize capital-dependence or history-dependence in their models (*Williams and Hastings, 2011*). Furthermore, models of reversal behavior in ecological settings generally rely upon the presence of exogenous environmental variation (*Jansen and Yoshimura, 1998*; *Roy et al., 2005*; *Benaïm and Schreiber, 2009*; *Wolf et al., 2005*; *Kussell and Leibler, 2005*; *Acar et al., 2008*; *Levine and Rees, 2004*). Without exogenous variation, the paradoxes do not occur. The broader applicability of Parrondo's paradox to ecological systems thus remains under-explored.

This lacuna remains despite the abundance of biological examples that exhibit history-dependent dynamics. The fitness of alleles may depend on the presence of genetic factors and epigenetic factors in previous generations (*Reed, 2007*). More generally, the fitness of any one gene can depend on the composition of other genes already present in a population, enabling the evolution of complex adaptations like multicellularity through ratcheting mechanisms (*Libby and Ratcliff, 2014*). Such mechanisms have recently been shown to help stabilize these complex adaptations (*Libby et al., 2016*). In ecological contexts, the storage effect can ensure that gains previously made in good years can promote persistence in less favorable times (*Warner and Chesson, 1985*; *Levine and Rees, 2004*). Species-induced habitat destruction or resource production can also have time-delayed effects on population growth, resulting in non-linear phenomena like punctuated evolution (*Yukalov et al., 2009*, *Yukalov et al., 2014*).

In this paper, we present a biologically feasible population model which exhibits counter-intuitive reversal behavior due to the presence of history-dependent and capital-dependent dynamics. Unlike most other studies, these dynamics do not rely upon the assumption of exogenous environmental variation. In our model, we consider a population that exists in two behaviorally distinct forms: as

nomads, and as colonists. Numerous organisms exhibit analogous behavioral diversity, from slime moulds (amoeba vs. plasmodia) (**Baldauf and Doolittle, 1997**) and dimorphic fungi (yeast vs. hyphae) (**Bastidas and Heitman, 2009**) to jellyfish (medusae vs. polyps) (**Lucas et al., 2012**) and human beings. One model organism which exhibits this sort of behavior, to which our study might apply, is the amoeba *Dictyostelium discoideum* (**Annesley and Fisher, 2009**).

Nomads live relatively independently, and thus are unaffected by either competition or co-operation. Under poor environmental conditions, they are subject to steady extinction. Colonists live in close proximity, and are thus subject to both competitive and co-operative effects. They may also deplete the resources of the habitat they reside in over time, resulting in long-term death. However, if these organisms are endowed with sensors that inform them of both population density and the state of the colonial habitat, they can use this information to switch from one behavior to another. Significantly, we find that an appropriate switching strategy paradoxically enables both population persistence and long-term growth – an ecological Parrondo's paradox.

## Population model

Two sub-populations comprise our model: the nomadic organisms, and the colonial ones. In a similar vein to habitat-patch models, organisms that exist in multiple sub-populations can be modelled as follows:

$$\frac{dn_i}{dt} = g_i(n_i) + \sum_j s_{ij} n_j - \sum_j s_{ji} n_i \tag{1}$$

where $n_i$ is the size of sub-population $i$, $g_i$ is the function describing the growth rate of $n_i$ in isolation, and $s_{ij}$ is the rate of switching to sub-population $i$ from sub-population $j$. Population sizes are assumed to be large enough that **Equation 1** adequately approximates the underlying stochasticity.

## Nomadism

Let $n_1$ be the nomadic population size. In the absence of behavioral switching, the nomadic growth rate is given by

$$g_1(n_1) = -r_1 n_1 \tag{2}$$

where $r_1$ is the nomadic growth constant. Nomadism is modelled as a losing strategy by setting $-r_1 < 0$, such that $n_1$ decays with time. In the context of Parrondo's paradox, nomadism corresponds to the 'agitating' strategy, or Game A. Importantly, competition among nomads, as well as between nomads and colonists, is taken to be insignificant, due to the independence of a nomadic lifestyle.

## Colonialism

Colonial population dynamics will be modelled by the well-known logistic equation, with carrying capacity $K$, but with two important modifications.

Firstly, the Allee effect is taken into account. This serves two roles: it captures the cooperative effects that occur among colonial organisms, and it ensures that the growth rate is negative when the population falls below a critical capacity $A$. Let $n_2$ be the colonial population size. In the absence of behavioral switching, the colonial growth rate is given by

$$g_2(n_2) = r_2 n_2 \left( \frac{n_2}{\min(A,K)} - 1 \right) \left( 1 - \frac{n_2}{K} \right) \tag{3}$$

where $r_2$ is the colonial growth constant. Setting $r_2 > 0$, we have a positive growth rate when $A < n_2 < K$, and a negative growth rate otherwise. The $\min(A,K)$ term ensures that when $K < A$, $g_2$ is always zero or negative, as would be expected.

Secondly, the carrying capacity $K$ changes at a rate dependent upon the colonial population size, $n_2$, accounting for the destruction of environmental resources over the long run.

The rate of change of $K$ with respect to $t$ is given by

$$\frac{dK}{dt} = \alpha - \beta n_2 \tag{4}$$

where $\alpha > 0$ is the default growth rate of $K$, and $\beta > 0$ is the per-organism rate of habitat destruction. An alternative interpretation of this equation is that $K$, the short-term carrying capacity, is dependent on some essential nutrient in the environment, and that this nutrient is slowly depleted over time at a rate proportional to $\beta n_2$.

Let $n^* = \frac{\alpha}{\beta}$, the critical population level at which no habitat destruction occurs. $\frac{dK}{dt}$ is zero when $n_2 = n^*$, positive when $n_2 < n^*$, and negative when $n_2 > n^*$. $n^*$ can thus also be interpreted as the long-term carrying capacity. Clearly, if the long-term carrying capacity $n^* < A$, the only stable point of the system becomes $n_2 = 0$. Under this condition, colonialism is a losing strategy as well.

Note that $g_2$ increases as $K$ increases, and that $K$ increases more quickly as $n_2$ decreases. In the context of Parrondo's paradox, colonialism can thus serve as a 'ratcheting' strategy, or Game B, because the rate of growth is implicitly dependent upon the colonial population in the past. Another way of understanding the 'ratcheting' behavior is through the lens of positive reactivity (*Williams and Hastings, 2011*; *Hastings, 2001*, *Hastings, 2004*). In the short-term, $n_2 = A$ is a positively reactive equilibrium, because small upwards perturbations of $n_2$ away from $A$ will result in rapid growth towards $K$ before a slow decrease back down towards $A$.

## Behavioral switching

Organisms are able to detect the amount of environmental resources available to them, and by proxy, the carrying capacity of the population. Thus, they can undergo behavioral changes in response to the current carrying capacity.

Here, we model organisms that switch to nomadic behavior from colonial behavior when the carrying capacity is low ($K < L_1$), and switch to colonial behavior from nomadic behavior when the carrying capacity is high ($K > L_2$), where $L_1 \leq L_2$ are the switching levels. Let $r_s > 0$ be the switching constant. Using the notation from *Equation 1*, switching rates can then be expressed as follows:

$$s_{12} = \begin{cases} r_s & \text{if } K < L_1 \\ 0 & \text{otherwise} \end{cases} \qquad s_{21} = \begin{cases} r_s & \text{if } K > L_2 \\ 0 & \text{otherwise} \end{cases} \tag{5}$$

A variety of mechanisms might trigger this switching behavior in biological systems. For example, since the nomadic organisms are highly mobile, they could frequently re-enter their original colonial habitat after leaving it, and thus be able to detect whether resource levels are high enough for recolonization. It should also be noted that the decision to switch need not always be 'rational' (i.e. result in a higher growth rate) for each individual. Switching behavior could be genetically programmed, such that 'involuntary' individual sacrifice ends up promoting the long-term survival of the species.

## Reduced parameters

Without loss of generality, we scale all parameters such that $\alpha = \beta = 1$. *Equation 4* thus becomes:

$$\frac{dK}{dt} = 1 - n_2 \tag{6}$$

Hence, $n^* = \frac{\alpha}{\beta} = 1$. All other population sizes and capacities can then be understood as ratios with respect to this critical population size. Additionally, since $\beta = 1$, $r_1$, $r_2$ and $r_s$ can be understood as ratios to the rate of habitat destruction. For example, if $r_2 \gg 1$, this means that colonial growth occurs much faster than habitat destruction. Time-scale separation between the population growth dynamics and the habitat change dynamics can thus be achieved by setting $r_1, r_2 \gg 1$. Similarly, the separation between the behavioral switching dynamics and the population growth dynamics can be achieved by setting $r_s \gg r_1, r_2$.

## Results

Simulation results revealed population dynamics that could be categorized into the following regimes:

1. Without behavioral switching ($r_s = 0$)
    a. Extinction for both sub-populations
    b. Extinction for nomadic organisms, survival for colonial organisms
2. With behavioral switching ($r_s > 0$)
    a. Extinction for both sub-populations
    b. Survival through periodic behavioral alternation
    c. Long-term growth through strategic alternation

Importantly, there were conditions under which both sub-populations would go extinct in the absence of behavioral switching (regime 1a), but collectively survive if behavioral switching was allowed (regime 2b), thereby exhibiting Parrondo's paradox. The following sections describe the listed regimes in greater detail, with a focus upon the regimes involved in the paradox. Figures generated via numerical simulation are provided as examples of behavior within each regime.

## Extinction in the absence of switching

As described earlier, both nomadic and colonial behaviors can be modelled as losing strategies given the appropriate parameters. Simulations across a range of parameters elucidated the conditions which resulted in extinction for both strategies. *Figure 1a* shows an example when both strategies are losing, resulting in extinction, while *Figure 1b* shows an example where only the colonial sub-population survives.

It is clear from *Equation 2* that the growth rate of the nomadic population $n_1$ is always negative, because of the restriction that $r_1 > 0$. Hence, nomadism is always a losing strategy.

However, the conditions under which colonial behavior is a losing strategy are more complicated. Complex dynamics occur when the critical capacity $A$ is just below 1 that can result in either survival or extinction. Nonetheless, it can be shown that when $A > 1$, extinction occurs (as in *Figure 1a*), and that survival is only possible when $A$ is significantly less than 1 (as in *Figure 1b*). That is:

$$A > 1 \quad \text{(colonial extinction guaranteed)} \tag{7}$$

$$A < 1 \quad \text{(colonial survival possible )} \tag{8}$$

The intuition behind this is straightforward. Suppose that initially, $A < n_2 < K$, so that the growth rate is positive. When $A < 1$, the colonial population $n_2$ increases until it reaches the carrying capacity $K$, following which they converge in tandem until stabilizing at the critical population size, $n_2 = K = 1$. However, when $A > 1$, $n_2 = K = 1$ is no longer a stable equilibrium, since $dn_2/dt < 0$ when $n_2 < A$, resulting in the eventual extinction of the population. For a formal proof, refer to Theorem A.3.

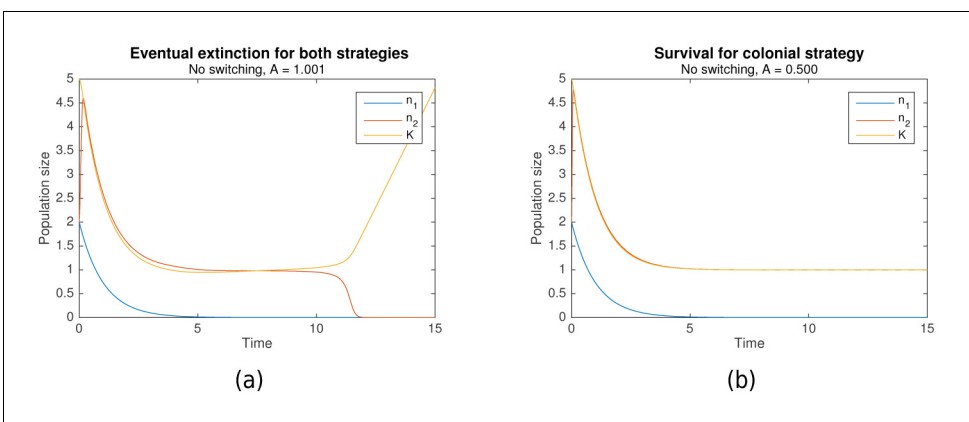

**Figure 1.** In the absence of switching, (a) eventual extinction for both strategies vs (b) survival for the colonial strategy. Initial conditions for both are $n_1 = 2$, $n_2 = 2$, $K = 5$. Shared parameters are $r_s = 0$, $r_1 = 1$, $r_2 = 10$. For (a), $A = 1.001$. For (b), $A = 0.5$.

## Survival through periodic alternation

We now restrict our analysis to the case where $A > 1$. Under this condition, both nomadism (Game A) and colonialism (Game B) are losing strategies when played individually. Paradoxically, it is possible to combine these two strategies through behavioral switching such that population survival is ensured, thereby producing an overall strategy that wins.

Simulation results over a range of parameters have predicted this paradoxical behavior, and also elucidated the conditions under which it occurs. *Figure 2a* is a typical example where the population becomes extinct, even though it undergoes behavioral switching, while *Figure 2b* is a typical example where behavioral switching ensures population survival.

Conceptually, this paradoxical survival is possible because the colonial strategy, or Game B, is history-dependent. In particular, the colonial growth rate $dn_2/dt$ is dependent upon the carrying capacity $K$, which in turn is dependent upon previous levels of $n_2$. Behavioral switching to a nomadic strategy decreases the colonial population size, allowing the resources in the colonial environment, represented by $K$, to recover. Switching back to a colonial strategy then allows the population to take advantage of the newly generated resources. Because switching occurs periodically, as can be seen in *Figure 2b*, it should be noted that the organisms need not even detect the amount of resources present in the environment to implement this strategy. A biological clock would be sufficient to trigger switching behavior.

The exact process by which survival is ensured can be understood by analysing the simulation results in detail. In the nomadic phase, the colonial population $n_2$ is close to zero, the nomadic population $n_1$ undergoes slow exponential decay, and the carrying capacity $K$ undergoes slow linear growth. $K$ increases until it reaches $L_2$, which triggers the switch to colonial behavior.

The population thus enters the colonial phase. If the colonial population $n_2$ exceeds the critical capacity $A$ at this point, then $n_2$ will grow until it slightly exceeds the carrying capacity $K$. Subsequently, $n_2$ decreases in the tandem with $K$ until $K$ drops to $L_1$, triggering the switch back to the nomadic phase. However, if $n_2 < A$ when the colonial phase begins, the colonial population goes extinct, as can be seen in *Figure 2a*. Hence, a basic condition for survival is that $n_2 \geq A$ at the start of each colonial phase.

This implies that, by the end of the nomadic phase, $n_1$ needs to be greater by a certain amount than $A$ as well. Otherwise, there will be insufficient nomads to form a colony which can overcome the Allee effect. Under the reasonable assumption that the rate of behavioral switching is much faster than either colonial or nomadic growth ($r_s \gg r_1, r_2$), it can be shown more precisely that at the end of the nomadic phase, $n_1$ needs to be greater than a critical level $B$, which is related to $A$ by the equation:

$$A = B - (1-B)W_0\left(\frac{B}{1-B}\exp\frac{B}{1-B}\right) \tag{9}$$

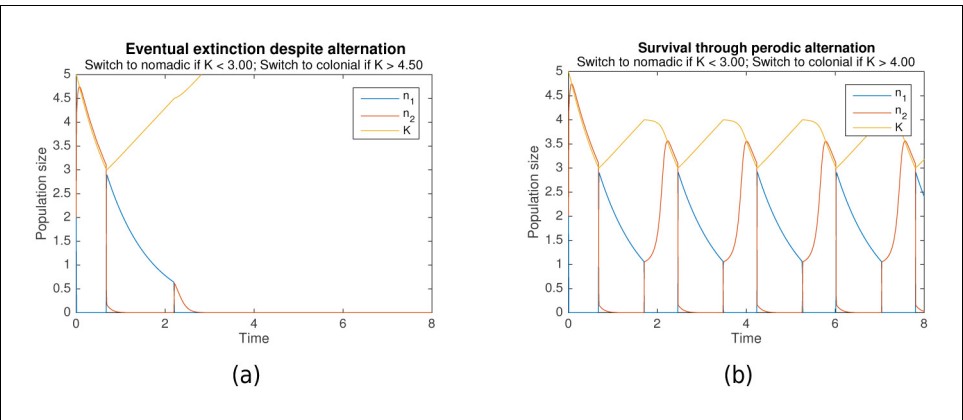

**Figure 2.** Behavioral switching which results in (a) extinction vs (b) survival. Initial conditions for both are $n_1 = 2$, $n_2 = 2$, $K = 5$. Shared parameters are $r_s = 1000$, $r_1 = 1$, $r_2 = 10$. For (a), $L_1 = 3$, $L_2 = 4.5$. For (b), $L_1 = 3$, $L_2 = 4$.

A full derivation is provided in the Appendix (Theorem A.4). Here, $W_0(x)$ is the principal branch of the Lambert W function. Qualitatively speaking, $B$ is a function of $A$ on the interval $(1,\infty)$ that increases in an exponential-like manner, and that approaches 1 when $A$ does as well. Thus, $B \geq A$, as expected.

The greater the difference between the switching levels, the longer the nomadic phase will last, because it takes more time for $K$ to increase to the requisite value for switching, $L_2$. And the longer the nomadic phase lasts, the more $n_1$ will decay. If, at the end of the nomadic phase, the value that $n_1$ decays to happens to be less than $B$, then the population will fail to survive. It follows that there should be some constraint on the difference between the switching levels $L_1$ and $L_2$.

Under the same assumption that $r_s \gg r_1, r_2$, such a constraint can be derived:

$$L_2 < L_1 + \frac{1}{r_1} \ln \frac{L_1 + W_0(-L_1 e^{-L_1})}{B} \tag{10}$$

Survival is ensured given the following additional condition:

$$\text{There exists } t^* \geq t_0 : n_2(t^*) = K(t^*) \geq L_1 \tag{11}$$

where $t_0$ marks the start of an arbitrary colonial phase, and $t^*$ marks the time of intersection between $n_2$ and $K$ during that phase. In other words, $n_2$ has to grow sufficiently quickly during the colonial phase such that it exceeds both $K$ and $L_1$ before switching begins. This can be seen occurring in *Figure 2b*. In accordance with intuition, numerical simulations predict that this occurs when the colonial growth constant is sufficiently large ($r_2 \gg r_1$), as can be seen in the Figures. (The Figures also show that $r_1$ close to 1, but this is not strictly necessary.) Collectively, *Equations 10–11* are sufficient conditions for population survival. Mathematical details are provided in the Appendix (Theorems A.5 and A.6).

Note that *Equation 10* contains an implicit lower bound on $L_1$. Since $L_2 \geq L_1$ by stipulation, we must have $\ln[L_1 + W_0(-L_1 e^{-L_1})] > \ln B$ for survival. The following bound is thus obtained:

$$L_1 > \frac{Be^B}{e^B - 1} \tag{12}$$

On the other hand, under the assumptions made, there is no upper bound for $L_1$, and hence no absolute upper bound for $L_2$ either. This suggests that given a sufficiently well-designed switching rule, $K$ can grow larger over time while ensuring population survival. Such a rule is investigated in the following section.

## Long-term growth through strategic alternation

Suppose that, in addition to being able to detect the colonial carrying capacity, nomads and colonists are able to detect or estimate their current population size. This might happen by proxy, by communication, or by built-in estimation of the time required for growth or decay to a certain population level. The following switching rule then becomes possible:

$$\begin{aligned} &\text{When } n_2 = K, \, dn_2/dt > 0, \text{ set } L_1 = K, \, L_2 = \infty \\ &\text{When } n_1 = B, \, dn_1/dt < 0, \text{ set } L_2 = K \end{aligned} \tag{13}$$

That is, $L_1$ is set to the carrying capacity $K$ whenever $n_2$ rises to $K$, resulting immediately in a switch to nomadic behavior, and that $L_2$ is in turn set to $K$ whenever $n_1$ falls to $B$, resulting in an immediate switch to colonial behavior.

This switching rule is optimal according to several criteria. Firstly, by switching to nomadic behavior just as $n_2$ reaches $K$, it ensures that $dn_2/dt \geq 0$ for the entirety of the colonial phase. As such, it avoids the later portion of the colonial phase where $K$ and $n_2$ decrease in tandem, and maximizes the ending value $n_2$. Consequently, it also maximizes the value of $n_1$ at the start of each nomadic phase.

Furthermore, by switching to colonial behavior right when $n_1$ decays to $B$, the rule maximizes the duration of the nomadic phase while ensuring survival. This in turn means that the growth of $K$ is maximized, since the longer the nomadic phase, the longer that $K$ is allowed to grow.

In fact, this switching rule is a paradigmatic example of how Parrondo's paradox can be achieved. It plays Game A, the nomadic strategy, for as long as possible, in order to maximize $K$ and hence the returns from Game B. And then it switches to Game B, the colonial strategy, only for as long as the returns are positive ($dn_2/dt > 0$), thereby using it as a kind of ratchet.

Suppose that $K$ grows more during each nomadic phase than it falls during each colonial phase. Then the switching rule is not just optimal, but it also enables long-term growth. Simulation results predict that this can indeed occur. *Figure 3a* shows long-term growth of $K$ from $t = 0$ to $t = 10$, while *Figure 3b* shows that with the same initial conditions, this continues until $t = 300$ with no signs of abating. Together with $K$, the per-phase maximal values of $n_1$ and $n_2$ increase as well.

In the cases shown, long-term growth is achieved because $K$ indeed grows more during each nomadic phase than it falls during the subsequent colonial phase. As can be seen from *Figure 3a*, this is, in turn, because the nomadic phase lasts much longer than the colonial phase, such that the amount of environmental destruction due to colonialism is limited. Simulation results predict that this generally occurs as long as the colonial growth rate is sufficiently large ($r_2 \gg r_1$).

An interesting phenomenon that can be observed from *Figure 3b* is how the nomadic population size $n_1$, which peaks at the start of each nomadic phase, eventually exceeds the carrying capacity $K$, and then continues doing so by increasing amounts at each peak. This is, in fact, a natural consequence of the population model. When $n_2$ grows large, the assumption that switching is much faster than colonial growth starts to break down. This occurs even though $r_s \gg r_2$, due to the increasing contribution of the $\left(\frac{n_2}{A} - 1\right)$ factor in *Equation 3*.

The result is that when a large colonial population begins switching to nomadism, a significant number of colonial offspring are simultaneously being produced. These offspring also end up switching to a nomadic strategy, resulting in more nomadic organisms than there were colonial organisms before. A particularly pronounced example of this is shown in *Figure 4*.

However, this same phenomenon also introduces a limiting behavior to the pattern of long-term growth. As *Figure 5* shows, when the same simulation as in *Figure 3a and b* is continued to $t = 1000$, peak levels of $n_1$, $n_2$ and $K$ eventually plateau around $t = 650$.

This occurs because sufficiently high levels of $n_2$ cause a qualitative change in the dynamics of behavioral switching. Normally, switching to nomadic behavior starts when $K$ falls below $L_1$, and ends when $K$ rises above it again. $K$ rises towards the end of the switch, when $n_2$ levels fall below the critical level of $n^* = 1$. But when $n_2$ is sufficiently large, the faster production of colonial offspring drags out the duration of switching, as seen in *Figure 4*. The higher levels of $n_2$, combined with the longer switching duration, causes an overall drop in $K$ by the end of the switching period. Because the increase in $K$ during the subsequent nomadic phase is unable to overcome this drop, $K$ stops increasing in the long-run.

Nonetheless, it is clear that significant long-term gains can be achieved via the optimal switching rule. Under the conditions of fast colonial growth and even faster switching ($r_s \gg r_2 \gg r_1 \simeq 1$, as in *Figures 3a–5*), these gains are several orders of magnitude larger than the initial population levels, a

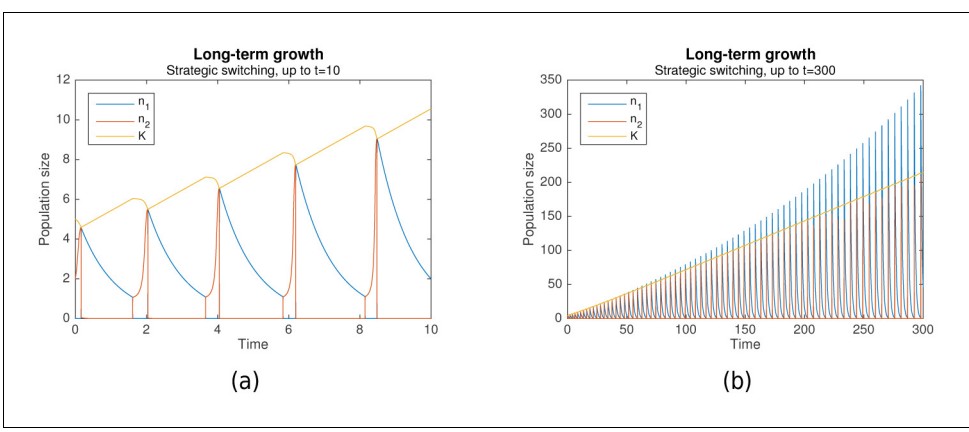

**Figure 3.** Long-term growth through strategic alternation. Initial conditions are $n_1 = 0$, $n_2 = 2$, $K = 5$. Parameters are $A = 1.001$, $L_1 = 3$, $L_2 = 4$, $r_s = 1000$, $r_1 = 1$, $r_2 = 10$.

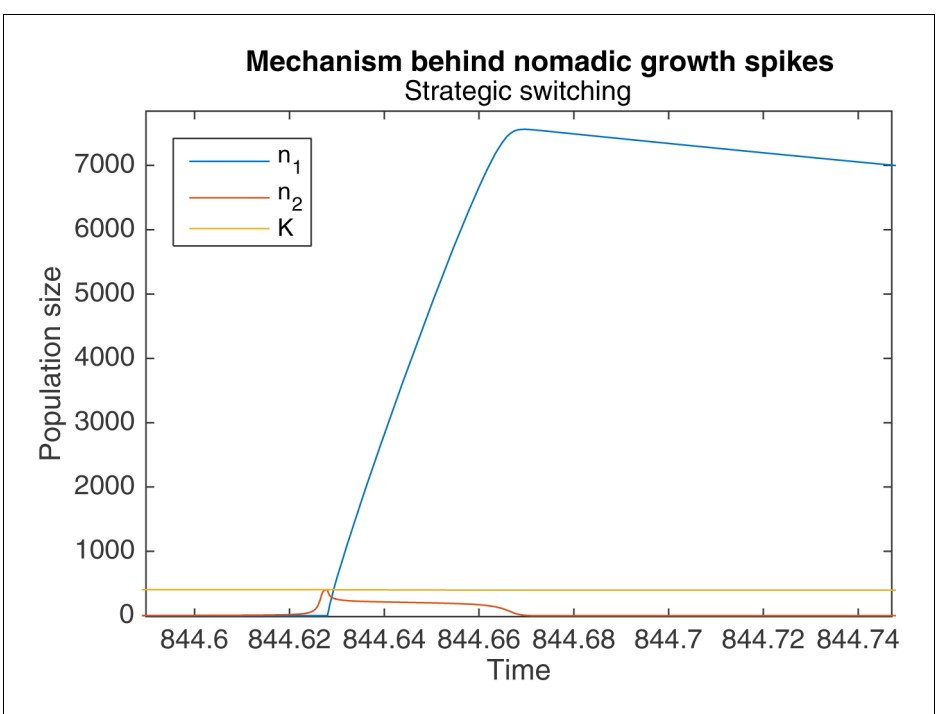

**Figure 4.** Zoomed-in portion of *Figure 5* that shows the mechanism behind the spikes in $n_1$. When $n_2$ is large, switching takes longer, causing a drop in $K$, and a large increase in $n_1$.

huge departure from the long-term extinction that occurs in purely colonial or nomadic populations.

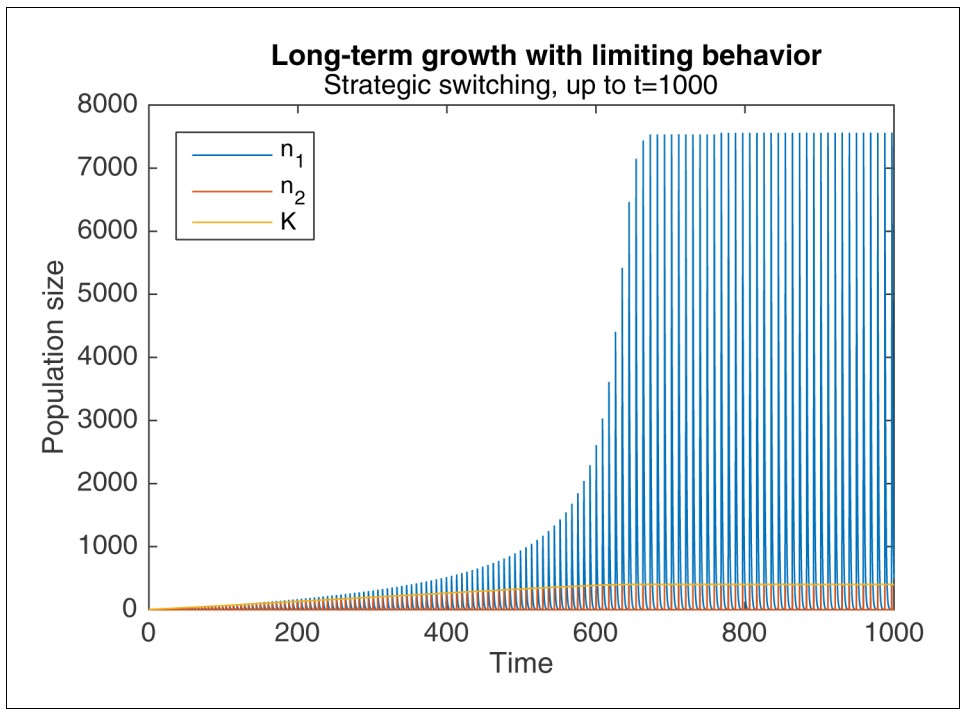

**Figure 5.** Long-term growth through strategic alternation, up to $t = 1000$. Initial conditions are $n_1 = 0$, $n_2 = 2$, $K = 5$. Parameters are $A = 1.001$, $r_s = 1000$, $r_1 = 1$, $r_2 = 10$.

Limiting behavior eventually emerges, but this is to be expected in any realistic biological system.

## Survival and growth under additional constraints

Our proposed model is convenient for the functional understanding of growth and survival, and can be easily modified for a variety of applications. Additional constraints can be imposed under which survival and long-term growth are still observed. For example, in many biological systems, the dynamics of habitat change might occur on a slower timescale than both colonial and nomadic growth (i.e. $r_1, r_2 \gg 1$). *Figure 6* shows the simulation results when this timescale separation exists ($r_1 = 10$, $r_2 = 100$ for (a), $r_1 = 100$, $r_2 = 1000$ for (b)). It can clearly be seen that survival is still possible under such conditions.

Another practical constraint that can be imposed is limiting the growth of the carrying capacity to some maximal value $K_{\max}$, capturing the fact that the resources in any one habitat do not grow infinitely large. This can be achieved by modifying *Equation 6* as follows:

$$\frac{dK}{dt} = (1 - n_2)\left(1 - \frac{K}{K_{\max}}\right) \tag{14}$$

*Figure 6* already takes this constraint into account, showing that survival through periodic alternation is achievable under both bounded carrying capacity and slow habitat change, as long as the maximum carrying capacity is sufficiently high ($K_{\max} = 20$). As *Figure 7* shows, even long-term growth is possible, under both fast habitat change (*Figure 7a*) and slow habitat change (*Figure 7b*). In both cases, the carrying capacity $K$ converges towards a maximum value as it approaches $K_{\max}$.

## Discussion

The results presented in this study demonstrate the theoretical possibility of Parrondo's paradox in an ecological context. Many evolutionary strategies correspond to the strategies that we have termed here as 'nomadism' and 'colonialism'. In particular, any growth model that is devoid of competitive or collaborative effects is readily captured by *Equation 2* (nomadism), while any logistic growth model which includes both the Allee effect and habitat destruction can be described using *Equations 3 and 4* (colonialism). Many organisms also exhibit behavioral change or phenotypic switching in response to changing environmental conditions. By incorporating this into our model, we have demonstrated that nomadic-colonial alternation can ensure the survival of a species, even when nomadism or colonialism alone would lead to extinction. Furthermore, it has been demonstrated that an optimal switching rule can lead to long-term population growth.

The switching rules which lead to survival and long-term growth are analogous to the periodic alternation between games that produces a winning expectation in Parrondo's paradox. If one views the carrying capacity $K$ as the capital of the population, then it is clear that *Equation 5* is a capital-

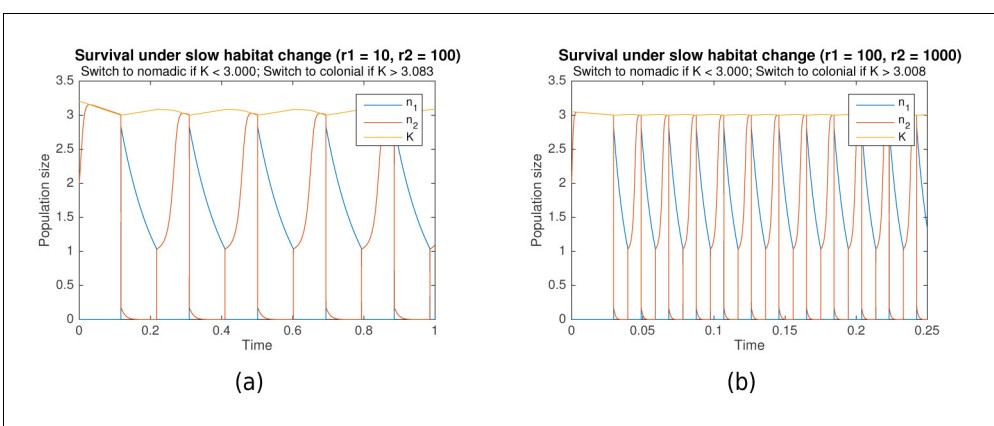

**Figure 6.** Survival through periodic alternation under slow habitat change. Parameters for (a) are $r_s = 10000$, $r_1 = 10$, $r_2 = 100$, $L_1 = 3$, $L_2 = 3.083$. For (b), $r_s = 100000$, $r_1 = 100$, $r_2 = 1000$, $L_1 = 3$, $L_2 = 3.008$. Shared parameters are $K_{max} = 20$, $A = 1.001$.

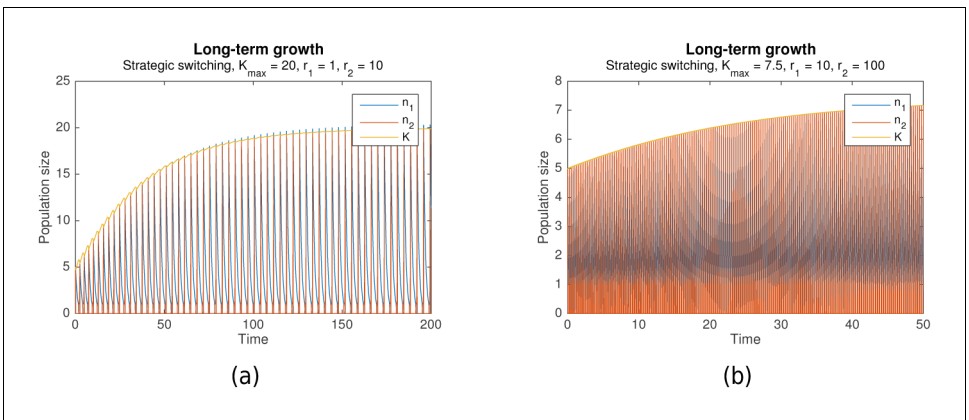

**Figure 7.** Long-term growth through strategic alternation with a bounded carrying capacity. Initial conditions are $n_1 = 0$, $n_2 = 2$, $K = 5$, shared parameters are $A = 1.001$. For (a), $K_{max} = 20$, $r_s = 1000$, $r_1 = 1$, $r_2 = 10$ (fast habitat change). For (b), $K_{max} = 7.5$, $r_s = 10000$, $r_1 = 10$, $r_2 = 100$ (slow habitat change).

dependent switching rule. By setting the appropriate amounts of capital at which switching should occur, survival and growth can be achieved. Survival is achieved by ensuring that Game A, or nomadism, is never played beyond the point where extinction is inevitable, that is, the point where $n_1$ falls below the critical level $B$. Long-term growth is additionally achieved by ensuring that Game B, or colonialism, is only played in the region where gains are positive, that is, when $A < n_2 < K$ such that $dn_2/dt > 0$. The history-dependent dynamics of Game B are thus optimally exploited.

Several limitations of the present study should be noted. Firstly, the study only focuses on cases where nomadism and colonialism are individually losing strategies, despite the abundance of similar strategies that do not lose in the real world. This is because assuming individually losing strategies in fact leads to a stronger result – if losing variants of nomadism and colonialism can be combined into a winning strategy, it follows that non-losing variants can be combined in a similar way too (see Theorem A.7 in the Appendix).

Secondly, the population model does not encompass all variants of qualitatively similar behavior. For example, many other equations can be used to model the Allee effect (*Boukal and Berec, 2002*). Nonetheless, our proposed model is general enough that it can be adapted for use with other equations and be expected to produce similar results. Even the presence of the Allee effect is not strictly necessary, since the colonial population might die off at low levels because of stochastic fluctuations, rather than because of the effect. Theorem A.7 in the Appendix also demonstrates that paradoxical behavior can occur even without the Allee effect causing long-term death of the colonial population.

Thirdly, though it is trivially the case that pure nomadism and pure colonialism cannot out-compete a behaviorally-switching population, a more complex analysis of the evolutionary stability of behavioral switching is beyond the scope of this paper. Finally, spatial dynamics are not accounted for in this study. Exploring such dynamics is a goal for future work.

## Materials and methods

Numerical simulations were performed using code written in MATLAB (*Source code 1*) that relied on the *ode23* ordinary differential equation (ODE) solver. *ode23* is an implementation of an explicit Runge-Kutta (2,3) pair of Bogacki and Shampine. Simulations were performed with both behavioral switching turned off ($r_s = 0$) and turned on ($r_s > 0$). The accuracy of the simulation was continually checked by repeating all results with more stringent tolerance levels, ensuring that the final simulated parameters did not change significantly (by less than 1%). Both the relative error tolerance and absolute error tolerance were determined to be $10^{-9}$.

In the case of complex switching rules like *Equation 13* that required modifying differential equation parameters at specific time points, the *Events* option of MATLAB's ODE solvers was used to

detect when these points occurred. After each detection, the parameters were automatically modified as per the switching rule, and the simulation continued with the new parameters.

Broad regimes of model behavior were observed by running simulations across a wide range of parameters and initial conditions. General trends and conditions observed within each regime were formalized analytically, the details of which can be found in the Appendix. In these derivations, reasonable assumptions were made in order to make the model analytically tractable. In particular, it was assumed that the rate of behavioral switching was much faster than the rates of either colonial or nomadic growth ($r_s \gg r_1, r_2$), and that colonial growth rates were in turn much faster than the rate of habitat destruction ($r_2 \gg 1$). Initial conditions corresponding to unstable equilibria (e.g. $n_2 = K = 1 < A$) were avoided as unrealistic.

## Conclusion

Our comprehensive model captures both capital and history-dependent dynamics within a realistic ecological setting, thereby exhibiting Parrondo's paradox without the need for exogenous environmental influences. The possibility of an ecological Parrondo's paradox has wide-ranging applications across the fields of ecology and population biology. Not only could it provide evolutionary insight into strategies analogous to nomadism, colonialism, and behavioral diversification, it potentially also explains why environmentally destructive species, such as *Homo sapiens*, can thrive and grow despite limited environmental resources. By providing a theoretical model under which such paradoxes occur, our approach may enable new insights into the evolution of cooperative colonies, as well as the conditions required for sustainable population growth.

## Additional information

### Funding
No external funding was received for this work.

### Author contributions
Zhi Xuan Tan, Conception and design of the study, Programmed the simulations and characterizations of the data, Analysis and interpretation of data, Drafting and revising the article; Kang Hao Cheong, Conception and design of the study, Analysis and interpretation of data, Drafting and revising the article, Supervised the research

### Author ORCIDs
Kang Hao Cheong (iD) https://orcid.org/0000-0002-4475-5451

### Decision letter and Author response
Decision letter https://doi.org/10.7554/eLife.21673.sa1
Author response https://doi.org/10.7554/eLife.21673.sa2

## Additional files

### Supplementary files
• Source code 1. MATLAB code used for the numerical simulation is attached as additional file. This contains all the functions used in the analyses presented in the paper. There are accompanying comments that document the parameters used in the model.

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

## Appendix 1

### Switching dynamics

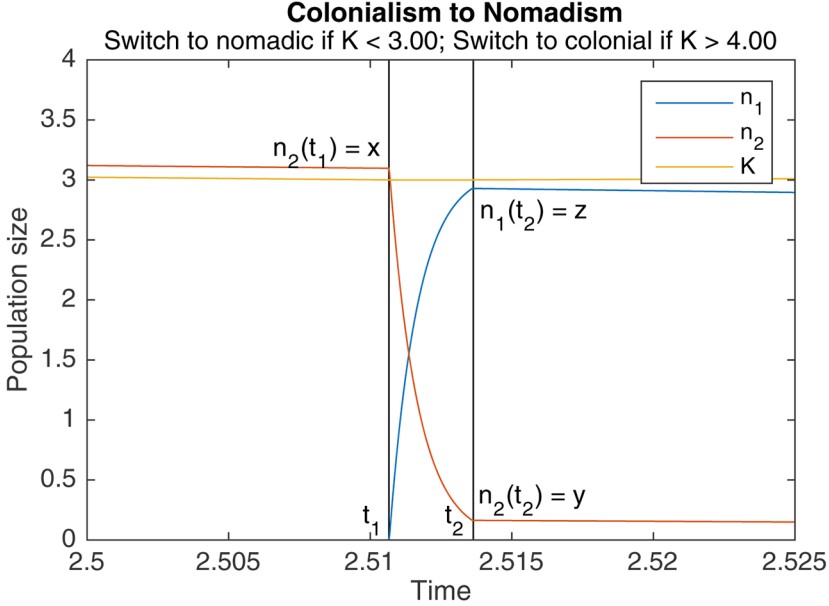

**Appendix 1—figure 1.** An example of switching from colonialism to nomadism. $t_1$ marks the start of the switch, $t_2$ marks the end. Other parameters are $A = 1.001$, $r_s = 1000$, $r_1 = 1$, $r_2 = 10$.

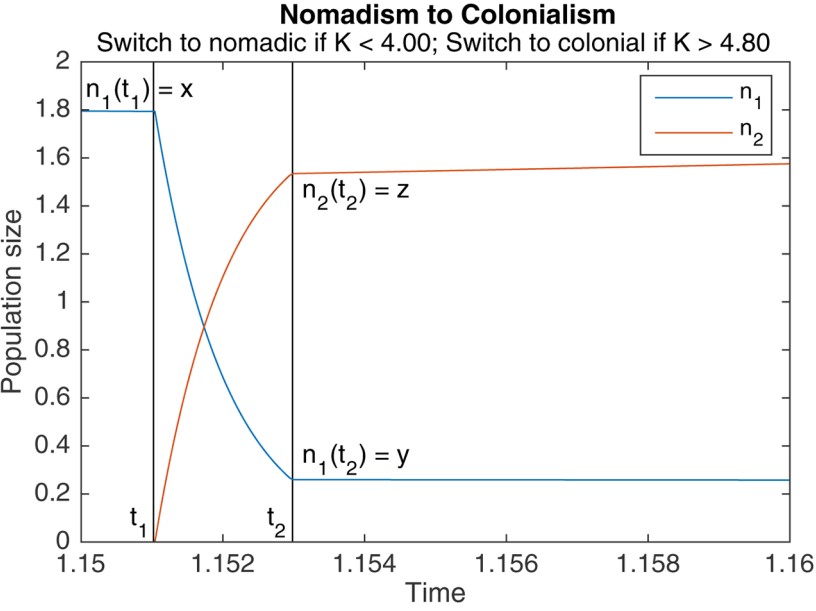

**Appendix 1—figure 2.** An example of switching from nomadism to colonialism. $t_1$ marks the start of the switch, $t_2$ marks the end. Other parameters are $A = 1.001$, $r_s = 1000$, $r_1 = 1$, $r_2 = 10$.

### Lemma A.1

Let $t_1$ and $t_2$ respectively be the start and end of a period of switching from sub-population $i$ to sub-population $j$. Then either of the following must hold:

$$\int_{t_1}^{t_2} 1 - n_2(t)\,dt = 0 \tag{A1}$$

$$n_i(t_2) = 0 \tag{A2}$$

Proof. Switching to nomadism begins when $K$ falls below $L_1$ and stops when it rises above $L_1$ again. Similarly, switching to colonialism begins when $K$ rises above $L_2$ and stops when it falls below $L_2$ again. In both cases then:

$$\Delta K = \int_{t_1}^{t_2} \frac{dK}{dt}\,dt = \int_{t_1}^{t_2} 1 - n_2(t)\,dt = 0$$

Alternatively, before $K$ rises or falls back to the appropriate level, $n_i$ may go extinct, prematurely ending the switch, in which case $n_i(t_2) = 0$.

## Theorem A.1 (Colonialism to nomadism)

Let $x$ be the colonial population at the end of a colonial phase, and $y$ be the colonial population at the start of the subsequent nomadic phase. Assuming that switching is much faster than colonial growth ($r_s n_2 \gg g_2(n_2)$), $y$ can be expressed in terms of $x_2$ as

$$y = -W_0(-xe^{-x}) \tag{A3}$$

where $W_0$ is the principal branch of the Lambert W function. Let $z$ be the *nomadic* population at the start of the subsequent nomadic phase. If switching is also much faster than nomadic growth ($r_s n_1 \gg g_1(n_1)$), and if the nomadic population is 0 before switching, we have

$$z = x + W_0(-xe^{-x}) \tag{A4}$$

Note that $z$ is an increasing function of $x$, and that $z$ converges to $x$ as $x$ grows larger.

Proof. Since $r_s n_2 \gg g_2(n_2)$, we have $dn_2/dt = -r_s n_2$. Hence, $n_2(t)$ decays exponentially from $x$ to $y$ over a duration of $\frac{1}{r_s}\ln(x/y)$. Substituting $n_2(t)$ into *Equation A1* and solving, we have

$$x - \ln x = y - \ln y$$

This means that $-x$ and $-y$ are both solutions to the equation $C = we^w$, where $C$ is some constant. The roots of this equation are given by the Lambert W function. Since $x > y$ and both are real numbers, $-x$ must lie on the $W_{-1}$ branch, while $-y$ lies on the principal $W_0$ branch. *Equation A3* follows.

For *Equation A4*, if $r_s n_1 \gg g_1(n_1)$, all the new nomadic organisms must have switched over from colonialism. It follows that $z = x - y$. This completes the proof.

## Theorem A.2 (Nomadism to colonialism)

Let $x$ be the nomadic population at the end of a nomadic phase, and $y$ be the nomadic population at the start of the subsequent colonial phase. Assuming that switching is much faster than both nomadic and colonial growth ($r_s n_1 \gg g_1(n_1), r_s n_2 \gg g_2(n_2)$ ), and that the colonial population is 0 before switching, $y$ can be expressed in terms of $x$ as

$$y = (1-x)W_0\left(\frac{x}{1-x}e^{\frac{x}{1-x}}\right) \tag{A5}$$

for $x > 1$. If $x \leq 1$, then $y = 0$, i.e., the nomads go extinct during switching. Let $z$ be the *colonial* population at the start of the subsequent colonial phase. We have

$$z = x - (1-x)W_0\left(\frac{x}{1-x}e^{\frac{x}{1-x}}\right) \tag{A6}$$

for $x > 1$, and $z = x$ otherwise. Note that in the first case, $z$ is an increasing function of $x$.

Proof. By our assumptions, we have $dn_1/dt = -r_s n_1$ and $dn_2/dt = r_s n_1$. Hence, $n_1(t)$ decays exponentially from $x$ to $y$ over a duration of $\frac{1}{r_s}\ln(x/y)$, and $n_2(t)$ increases correspondingly from 0 to $(x - y)$. Substituting $n_2(t)$ into *Equation A1* and solving, we have

$$(1 - x)(\ln x - \ln y) + (x - y) = 0$$

Suppose that $x > 1$, $y$ can be expressed in terms of $x$ using the principal branch of the Lambert W function, and *Equation A5* follows. If $x \leq 1$, then there is no way to satisfy *Equation A1* because $n_2$ will not exceed 1 during the switching period. It follows that $n_1$ must go extinct, in which case $y = 0$. The results for $z$ follow since $z = x - y$. This completes the proof.

## Extinction and survival

### Theorem A.3 (Colonists go extinct when $A > 1$)

When $A > 1$, pure colonialism leads to extinction in the long run ($\lim_{t \to \infty} n_2(t) = 0$), assuming initial conditions are not set to the unstable equilibrium $n_2 = K = 1 < A$.

Proof. There are three possible equilibria for $n_2$, at which $dn_2/dt = 0$: $n_2 = 0$, $n_2 = A$, and $n_2 = K$ (refer to *Equation 3*). However, when $n_2 = A > 1$ or $n_2 = K > 1$, we have $dK/dt = 1 - n_2 < 0$. $K$ decreases until $n_2 < K$, which implies $dn_2/dt < 0$. $n_2$ then decreases until $n_2 < A$, which implies convergence to 0 as $t \to \infty$.

Similarly, if $n_2 = K < 1$, then $dK/dt > 0$, such that $n_2 < K < 1 < A$ an infinitesimal moment later, which then leads to extinction. This leaves only $n_2 = 0$ and $n_2 = K = 1$ as equilibria, the latter of which is unstable to small perturbations of $n_2$ towards zero. This completes the proof.

### Theorem A.4 (Survival imposes a lower bound on $n_1$)

For survival to occur, $n_1(t_N) > B$ is a necessary condition, where $t_N$ denotes the end of any nomadic phase, and $B$ is a critical level related to $A$:

$$A = B - (1 - B)W_0\left(\frac{B}{1 - B}\exp\frac{B}{1 - B}\right) \tag{A7}$$

Proof. Let $t_C$ denote the start of the subsequent colonial phase. We know that $n_2(t_C) > A$ is required for the new colony to survive. $B$ is defined using *Equation A6* such that $n_2(t_C) = A$ if $n_1(t_N) = B$. Since $n_2(t_C)$ is an increasing function of $n_1(t_N)$, it follows that $n_1(t_N) > B$ in order for $n_2(t_C) > A$. This completes the proof.

### Theorem A.5 (Survival imposes constraints on $L_1$ and $L_2$)

Under the reasonable assumptions that switching is much faster than colonial or nomadic growth ($r_s n_1 \gg g_1(n_1), r_s n_2 \gg g_2(n_2)$), and that the difference between $n_2$ and $K$ is negligibly small once $n_2$ grows to exceed $K$ during the colonial phase, the following constraint on $L_1$ and $L_2$ is a necessary condition for survival through periodic alternation:

$$L_2 \leq L_1 + \frac{1}{r_1}\ln\frac{L_1 + W_0(-L_1 e^{-L_1})}{B} \tag{A8}$$

This also implies a lower bound on $L_1$, given by:

$$L_1 \geq \frac{Be^B}{e^B - 1} \tag{A9}$$

Proof. Consider a sequence of consecutive phases which we will term as CP$_1$, NP, CP$_2$, where CP$_1$ and CP$_2$ are colonial phases and NP is the nomadic phase between them. Since the nomadic phase NP is characterized by exponential decay, it is clear that $n_1(\text{NP-start}) = n_1(\text{NP-end})e^{r_1 T}$, where

$$T := \text{NP-end} - \text{NP-start} = \frac{1}{r_1}\ln\frac{n_1(\text{NP-start})}{n_1(\textit{NP-end})}$$

is the duration of the nomadic phase. During this period, the colonial population $n_2$ is close to or equal to 0. Since $dK/dt = 1 - n_2 \leq 1$, it follows that

$$K(\text{NP-end}) - K(\text{NP-start}) \leq 1 \cdot (\text{NP-end} - \text{NP-start}) = T$$

Note that $K(\text{NP-start}) = L_1$ and $K(\text{NP-end}) = L_2$ by definition. Combining this with the two equations above, we obtain

$$L_2 \leq L_1 + \frac{1}{r_1} \ln \frac{n_1(\text{NP-start})}{n_1(\text{NP-end})} \tag{A10}$$

For survival to occur, the colonial population at the start of CP$_2$ needs to be greater than $A$. By Theorem A.4, this implies that $n_1(\text{NP-end}) > B$. Substituting into the above, we have

$$L_2 < L_1 + \frac{1}{r_1} \ln \frac{n_1(\text{NP-start})}{B} \tag{A11}$$

Notice that by Theorem A.1, $n_1(\text{NP-start})$ is a function of $n_2(\text{CP}_1\text{-end})$, the colonial population at the end of CP$_1$. If CP$_1$ lasts long enough that $n_2$ exceeds $K$, then by assumption we have $n_2(\text{CP}_1\text{-end}) \simeq K(\text{CP}_1\text{-end})$, otherwise we have $n_2(\text{CP}_1\text{-end}) < K(\text{CP}_1\text{-end})$. Note that $K(\text{CP}_1\text{-end}) = L_1$. Thus, $n_2(\text{CP}_1\text{-end}) \leq L_1$. Then by *Equation A4* in Theorem A.1, we have $n_1(\text{NP-start}) \leq L_1 + W_0(-L_1 e^{-L_1})$. Substituting this into *Equation A11* gives us *Equation A8*, as desired.

Finally, recall that $L_2 > L_1$ by definition. It thus needs to be the case that

$$\ln \frac{L_1 + W_0(-L_1 e^{-L_1})}{B} > 0$$

Solving this gives us the lower bound on $L_1$. This completes the proof.

## Theorem A.6 (Survival is ensured if $n_2$ converges to $K$ quickly)

Under the reasonable assumption that $n_2 \simeq 0$ during each nomadic phase, then together with *Equation A8* as well as the assumptions made in A.5, the following constraint is sufficient to ensure survival:

$$\text{There exists } t^* \geq t_0 : n_2(t^*) = K(t^*) \geq L_1 \tag{A12}$$

where $t_0$ marks the start of an arbitrary colonial phase, and $t^*$ marks the time of intersection between $n_2$ and $K$ during that phase. In other words, $n_2$ has to grow sufficiently quickly during the colonial phase such that it exceeds both $K$ and $L_1$ before switching begins.

Proof. Let CP$_1$ denote the colonial phase in question. Given *Equation A12*, all switching to the nomadic phase only happens after $n_2$ exceeds $K$ during CP$_1$. We have already presumed in Theorem A.5 that once $n_2$ grows to exceed $K$, we have $n_2 - K \ll 1$. This implies that $n_2(\text{CP}_1\text{-end}) \simeq K(\text{CP}_1\text{-end}) = L_1$, which in turn implies that $n_1(\text{NP-start}) \simeq L_1 + W_0(-L_1 e^{-L_1})$ by Theorem A.1.

Given the new assumption that $n_2 \simeq 0$ during each nomadic phase, any value of $L_2$ that satisfies *Equation A10* will be *sufficient* for survival. Substituting $n_1(\text{NP-end}) > B$ along with our new expression for $n_1(\text{NP-start})$ into *Equation A10* retains this property, which means that any value of $L_2$ that satisfies *Equation A8* is sufficient for survival (as long as $L_2 > L_1$). Thus, given our assumptions, *Equation A8* and *Equation A12* collectively ensure survival. This completes the proof.

## Theorem A.7 (Behavioral alternation improves survival levels)

Suppose that pure nomadism or pure colonialism (or both) result in population survival. That is,

$$n_{i,\text{pure}}(\infty) := \lim_{t \to \infty} n_{i,\text{pure}}(t) > 0 \tag{A13}$$

where $i = 1$ or 2 or both. This can happen if $r_1 = 0$ (for nomadism) or $A < 1$ (for colonialism). Then there are conditions under which behavioral alternation between the two pure strategies will result in a

total population size with higher long-term periodic maxima. That is, given the appropriate conditions, then for any $t \geq 0$, there exists $t^* \geq t$ such that

$$n_1(t^*) + n_2(t^*) > n_{i,\mathrm{pure}}(\infty) \tag{A14}$$

for both $i = 1$ and $i = 2$.

Proof. As shown in the main text, there exist cases where periodic alternation between the levels $K = L_1$ and $K = L_2$ result in survival. Furthermore, there exists cases where $n_2(t^*) = K(t^*) \geq L_1$ for some $t^*$ during each colonial phase. For any such case where both nomadism and colonialism are losing strategies, notice that we would still have $n_2(t^*) = K(t^*) \geq L_1$ even if nomadism or colonialism were modified so that they did not lose.

To be precise, if $r_1 = 0$ (nomadism does not lose), then there would be no decay over each nomadic phase, which means that in the subsequent colonial phase, $n_2$ would start out higher and grow to reach $K$ even more quickly. If $A < 1$ (colonialism does not lose), then $g_2(n_2)$ would be even larger in magnitude, so $n_2$ would grow towards $K$ more quickly as well. In both cases, $K$ would decay less than it would have originally by the time $n_2(t^*) = K(t^*)$, so we would still have $n_2(t^*) > L_1$. It follows that as long as $L_1 > n_{i,\mathrm{pure}}(\infty)$, *Equation A14* will always be true. Such cases can be easily found for $i = 1$ (ensuring $L_1 > n_{1,pure}(0)$) and for $i = 2$ (ensuring $L_1 > n_{2,pure}(\infty) = 1$). This completes the proof.

