## [Decision Letter]

Thank you for submitting your article "Nomadic-colonial alternation can enable population growth despite habitat destruction: An ecological Parrondo's paradox" for consideration by *eLife*. Your article has been favorably evaluated by Ian Baldwin (Senior Editor) and three reviewers, one of whom is a member of our Board of Reviewing Editors. The reviewers have opted to remain anonymous.

The reviewers have discussed the reviews with one another and the Reviewing Editor has drafted this decision to help you prepare a revised submission.

All reviewers agree that this is an interesting paper that introduces Parrondo's paradox to an ecological setting. In this setting, the paradox consists of the fact that even though each of two sub-populations go extinct when left alone, migration between the sub-populations can lead to persistence. The mechanism enabling persistence is that one of the habitats can regenerate itself while the population spends time declining in the other habitat, so that after some time the regenerated habitat can harbor a growing population again for some time, before that habitat deteriorates again, and so on. I think this phenomenon is worth being brought to the attention of ecologists, and will likely enhance the conceptual toolbox of people concerned with conservation issues.

As you will see the reviewers have raised a number of concerns about the paper. Chief among those is the biological realism and relevance of some of the model assumptions and interpretations. These concerns need to be addressed in a major revision. After we have received the revision, we will make a decision as to the suitability of your paper for *eLife* based on how each of the following points were addressed:

1) There were concerns about the realism of some of the modelling assumptions. Specifically, the paper assumes that the carrying capacity in the habitat of the colonizers changes on the same rapid ecological time scale as the population density itself changes. This seems unrealistic. Is it possible to formulate an alternative model, e.g. with a time scale separation between population dynamics and the dynamics of the carrying capacity? If not, what is the rationale behind assuming the same time scales for those two dynamics?

2) In addition, the assumptions about the dynamics of the carrying capacity imply that in the absence of colonizers (i.e., if the population density of the colonizers is 0), the carrying capacity increases without bounds. Clearly, this is not realistic, and the question is whether it is possible to obtain the same results with more realistic assumptions, which would e.g. ensure that the carrying capacity saturates at some level even in the absence of colonizers.

3) As referee 1 points out, the assumed switching behavior may also not be very realistic. At the very least, this kind of switching behavior needs to be justified biologically. Are there other, biologically more relevant switching behaviors that would lead to the same results?

4) In general, all other reviewer comments and concerns need to be addressed in a satisfactory manner before a final decision regarding publication can be made.

*Reviewer #1:*

This is an interesting paper that introduces Parrondo's paradox to an ecological setting. In this setting, the paradox consists of the fact that even though each of two subpopulations go extinct when left alone, migration between the subpopulations can lead to persistence. The mechanism enabling persistence is that one of the habitats can regenerate itself while the population spends time declining in the other habitat, so that after some time the regenerated habitat can harbour a growing population again for some time, before that habitat deteriorates again, and so on. I think this phenomenon is worth being brought to the attention of ecologists, and will e.g. enhance the conceptual toolbox of people concerned with conservation issues.

The paper appears to be well executed technically, but I do have some concerns regarding the biological significance of the work. Chief among those is the fact that the carrying capacity is assumed to change on the same time scale as the population density (i.e., the "t" in eqs (1) and (4) is the same). On intuitive grounds, I would expect that the dynamics of the carrying capacity to be much slower than the population dynamics. Also, in the absence of any population in the habitat in which the carrying capacity is changing, the carrying capacity will grow without bounds according to eq. (4), which is obviously unrealistic. I think one could obtain a biologically more reasonable model by assuming some sort of time scale separation between carrying capacity dynamics and population dynamics (e.g. by assuming that \α and \β in equation (4) are <<1), and by assuming that the carrying capacity saturates as the population size in the habitat goes to 0. The question would then be whether the phenomenon observed by the authors would still be present under such conditions.

I am also not really convinced by the proposed switching behaviour (subsection "Behavioral switching" under "Population model"). The authors purpose that individuals switch from nomadic to colonial behaviour when the carrying capacity for the colonial dynamics is high enough. But how would nomadic individuals be able to assess the carrying capacity for the colonial life style (I can understand how the colonial individuals would assess that carrying capacity)? The more elaborate switching rule given by (13) is even less intuitive: why would colonial individuals switch to monadic life style once they reach carrying capacity? After all, this means that individuals trade a per capita growth rate of 0 (because they treat carrying capacity) with a negative per capita growth rate (as is the case for nomadic life style by assumption). I think it would make more sense to switch once the colonial growth rate is "negative enough", e.g. more negative than the (constant) negative per capita growth rate on the nomadic life style. Again, the question then becomes whether persistence can still be observed under such more realistic assumptions.

Overall, I think the paper is interesting, but needs a much more careful treatment of the biological realism underlying various assumptions, and as a consequence, a much more careful discussion of the biological relevance of the results obtained.

*Reviewer #2:*

This is a cleverly laid out paper that will introduce a relatively new perspective to ecological and evolutionary biologists. That is, this paper lays out an example of how switching "colonial and nomadic" behavior can lead to persistence despite each singular case not allowing persistence. While I think the general idea of organisms responding to variation in a manner that yields persistence or coexistence is far from new to ecologists and evolutionary ecologists, the perspective put forth is interesting and novel. My guess is that there are numerous existing models that may, in hindsight and with a little work, fit into this "capital" or "history dependent" version of Parrondo's paradox. I do think people studying complex systems may not be overwhelmingly surprised by the notion that an overall average growth rate less than zero in two strategies, or two of whatever (say competitors), can respond to variation (environmentally driven or internally driven) in a manner that yields persistence or coexistence but, again, the paper framed within the Parrondo's paradox seems potentially profitable for understanding and looking for this type of persistence outcome.

I have no major comments and I think this paper is well written, well analyzed (at least for the ideas laid out), and so very close to publishable as is.

*Reviewer #3*

General Assessment:

The manuscript under review ("Nomadic-colonial alternation can enable population growth despite habitat destruction: An ecological Parrondo's paradox") presents and analyzes a novel model for a population where individuals are one of two behavioral types, which are modelled as subpopulations in an ODE model. Colonial types cooperate, compete and (over the long term) reduce their own carrying capacity. Nomads slowly die off (in the parameter space analyzed). Offspring of each type can "switch" to the other and in the model they do so based on the carrying capacity of the colonial sub-population.

Through a careful and detailed analysis, the paper shows that the model demonstrates a paradoxical behavior: in situations where pure colonial or nomadic strategies would die out, switching can result in persistence. Such results are well known in ecology and evolution, but most often occur due to environmental variation [1]. Here, the results occur not because of environmental variation but due to the carrying-capacity-dependent switching rule modeled. This provides a closer analog to Parrondo games than in many ecological studies, which (where it is biologically realistic) might help unify various studies of such reversal behaviors (as called for in [1]).

The particular situation studied in the model of alternation between nomadic and colonial types is of great evolutionary interest as it mirrors both the evolution of cooperative colonies from modular organisms and even the evolution of multicellularity itself. Indeed, the language of ratcheting has been invoked in describing the latter process [2]. More recently, that same group has examined the stabilization of multicellularity through ratcheting in a genetic model ([3] which should be cited and discussed in any revision).

The model in the manuscript under review, although not explicitly evolutionary, provides an interesting avenue through which these questions might be explored in the future. However, the behavior demonstrated rests on some peculiarities of the model and its assumptions.

Concerns:

Although promising, the model as developed displays some strange and possibly unrealistic behavior, which may be due to atypical assumptions. In particular, modelling the carrying capacity K as a function of the population itself (eqn 3) in a single-species model is not common in ecology (e.g., no such model appears in Table 1 of [4]). One reason may be that, as seen here, K will diverge to infinity when populations become small (e.g., if the colonists die off; Figure 1A). The dynamics of K also play a role in the strange behavior explored in Figures 3 and 4, where a vast overabundance of nomads is produced during a long-lived period of switching, during which K is reduced. Because the switching dynamics (eqn 5) are based on the dynamics of K, and key to driving the analyzed behavior the generality of these behaviors is tied to the reasonableness and generality of this assumption.

[1] Paul David Williams and Alan Hastings. Paradoxical persistence through mixed-system dynamics: towards a unified perspective of reversal behaviours in evolutionary ecology. Proceedings of the Royal Society of London B: Biological Sciences, page rspb20102074, 2011.

[2] Eric Libby and William C. Ratcliff. 346(6208):426-427, 2014.

[3] Libby E, Conlin PL, Kerr B, Ratcliff WC. 2016 Stabilizing multicellularity through ratcheting. Phil. Trans. R. Soc. B 371: 20150444. http://dx.doi.org/10.1098/rstb.2015.0444

[4] Abbot, Ives. 2012 Single Species Population Models in "Encyclopedia of Theoretical Ecology" Hastings, Gross, Eds. UC Press.

---

## [Author Response]

[…] As you will see the reviewers have raised a number of concerns about the paper. Chief among those is the biological realism and relevance of some of the model assumptions and interpretations. These concerns need to be addressed in a major revision. After we have received the revision, we will make a decision as to the suitability of your paper for eLife based on how each of the following points were addressed:1) There were concerns about the realism of some of the modelling assumptions. Specifically, the paper assumes that the carrying capacity in the habitat of the colonizers changes on the same rapid ecological time scale as the population density itself changes. This seems unrealistic. Is it possible to formulate an alternative model, e.g. with a time scale separation between population dynamics and the dynamics of the carrying capacity? If not, what is the rationale behind assuming the same time scales for those two dynamics?

We started with a simpler population model for the functional understanding of growth and survival, and ensured that this model could be easily modified for different applications. We have now added a new subsection “Survival and Growth under Additional Constraints”, implementing all the constraints suggested by the reviewers. In particular, subsection "Reduced parameters" and subsection "Survival and growth under additional constraints", first paragraph address how time scale separation between the population dynamics and the dynamics of the carrying capacity can be achieved by modifying the rate parameters of the existing model. We note that there are many species that exhibit nomadic-like behavior with growth dynamics that could easily occur on the same time frame as habitat change. The spores of fungi and slime molds are one such example. Nonetheless, to demonstrate the generalizability of our model, we performed new simulations using r_2_ >> 1 (r_2_ = 100), and r_1_ >> 1 (r_1_ = 10), such that the rate of both colonial growth and nomadic decay were at least an order of magnitude faster than the rate of habitat destruction. The corresponding results are shown in subsection "Survival and growth under additional constraints" as well. Similar results were also obtained for r_1_ = 100 and r_2_ = 1000. As can be seen from the results, both survival and long-term growth are still possible under the suggested time scale separation. These new results would satisfactorily address the concerns of the reviewers.

2) In addition, the assumptions about the dynamics of the carrying capacity imply that in the absence of colonizers (i.e., if the population density of the colonizers is 0), the carrying capacity increases without bounds. Clearly, this is not realistic, and the question is whether it is possible to obtain the same results with more realistic assumptions, which would e.g. ensure that the carrying capacity saturates at some level even in the absence of colonizers.

Yes, similar results can be obtained even without the carrying capacity increasing without bounds. Using a logistic-like equation (Equation 14), we performed additional simulations that limited the growth of the carrying capacity to some maximal value *K*_max_. Please refer to the new subsection "Survival and growth under additional constraints" and Figures 6 and 7. For sufficiently high values of *K*_max_, both survival and long-term growth remain possible.

3) As referee 1 points out, the assumed switching behavior may also not be very realistic. At the very least, this kind of switching behavior needs to be justified biologically. Are there other, biologically more relevant switching behaviors that would lead to the same results?

Response: A variety of mechanisms might trigger the assumed switching behavior in biological systems. For example, since the nomadic organisms are highly mobile, they could frequently re-enter their original colonial habitat after leaving it, and thus be able to detect whether resource levels are high enough for recolonization. Alternatively, in the case of survival through periodic alternation, a biological clock would be sufficient to implement the periodic switching strategy, eschewing the need to detect resource levels altogether. It should also be noted that the decision to switch need not always be 'rational' (i.e. result in a higher growth rate) for each individual. Switching behavior could instead be genetically programmed (hard-wired), such that 'involuntary' individual sacrifice ends up promoting the long- term survival of the species. How this sort of 'selfless' behavior might have evolved is beyond the scope of our study. We simply note that sacrificial behavior is both possible and common in nature, as in the case of cellular slime molds which sacrifice themselves when collectively forming a fruiting body. Subsection "Behavioral switching", last paragraph and subsection "Survival through periodic alternation", third paragraph now include our justification.

4) In general, all other reviewer comments and concerns need to be addressed in a satisfactory manner before a final decision regarding publication can be made.

*Reviewer #1:*

[…] The paper appears to be well executed technically, but I do have some concerns regarding the biological significance of the work. Chief among those is the fact that the carrying capacity is assumed to change on the same time scale as the population density (i.e., the "t" in eqs (1) and (4) is the same). On intuitive grounds, I would expect that the dynamics of the carrying capacity to be much slower than the population dynamics.

As noted above, time scale separation between the population dynamics and the dynamics of the carrying capacity can be achieved by modifying the rate parameters of the existing model. Specifically, this can be achieved by setting r_1_ >> 1 and r_2_ >> 1, such that colonial growth and nomadic decay occur much more rapidly than habitat change. Subsection "Reduced parameters" and subsection "Survival and growth under additional constraints", first paragraph now address this point. As can be seen from the new results in subsection "Survival and growth under additional constraints", both survival and long-term growth are still possible under the suggested time scale separation.

Also, in the absence of any population in the habitat in which the carrying capacity is changing, the carrying capacity will grow without bounds according to eq. (4), which is obviously unrealistic. I think one could obtain a biologically more reasonable model by assuming some sort of time scale separation between carrying capacity dynamics and population dynamics (e.g. by assuming that \α and \β in equation (4) are <<1), and by assuming that the carrying capacity saturates as the population size in the habitat goes to 0. The question would then be whether the phenomenon observed by the authors would still be present under such conditions.

We thank reviewer #1 for the suggestions, which we have since taken into account in subsection "Survival and growth under additional constraints". Time scale separation has already been addressed above. Saturation of the carrying capacity was implemented by modifying Equation 6 (the reduced version of Equation 4) to incorporate logistic-like behavior, as stated in Equation 14. This limits the growth of the carrying capacity to a maximal value *K*_max_, capturing the fact that the resources do not grow infinitely large.

Figure 6 takes this constraint into account, showing that survival through periodic alternation is achievable under both bounded carrying capacity and slow habitat change, as long as the maximum carrying capacity is sufficiently high (*K*_max_ = 20). As Figure 7 shows, long-term growth is also possible under these constraints, with the carrying capacity K converging towards a maximum value as it approaches *K*_max_. Please refer to subsection "Survival and growth under additional constraints" for a full description of these results.

I am also not really convinced by the proposed switching behaviour (subsection "Behavioral switching"). The authors purpose that individuals switch from nomadic to colonial behaviour when the carrying capacity for the colonial dynamics is high enough. But how would nomadic individuals be able to assess the carrying capacity for the colonial life style (I can understand how the colonial individuals would assess that carrying capacity)?

As noted above, a variety of mechanisms might trigger the assumed switching behavior. For example, since the nomadic organisms are highly mobile, they could frequently re-enter their original colonial habitat after leaving it, and thus be able to detect whether resource levels are high enough for recolonization. This is now explained in subsection "Behavioral switching" , last paragraph.

Alternatively, in the case of survival through periodic alternation, a biological clock would be sufficient to implement the periodic switching strategy, eschewing the need to detect resource levels altogether. This is now discussed in subsection "Survival through periodic alternation", third paragraph. Both colonial and nomadic organisms might simply be ‘hard-coded’ to switch behaviors after a certain amount of time, generating periodic behavior that synchronizes with the regeneration of the colonial environment. This could well explain the existence of multi-form lifecycles like that of jellyfish, which alternate between colonial polyps and nomadic medusa.

The more elaborate switching rule given by (13) is even less intuitive: why would colonial individuals switch to monadic life style once they reach carrying capacity? After all, this means that individuals trade a per capita growth rate of 0 (because they treat carrying capacity) with a negative per capita growth rate (as is the case for nomadic life style by assumption). I think it would make more sense to switch once the colonial growth rate is "negative enough", e.g. more negative than the (constant) negative per capita growth rate on the nomadic life style. Again, the question then becomes whether persistence can still be observed under such more realistic assumptions.

We thank reviewer #1 for bringing up this concern. As noted earlier, this concern is only valid if we consider the colonial organisms to be selfish 'rational agents' which never choose to take a loss. This is now addressed in subsection "Behavioral switching", last paragraph. It is common knowledge that seemingly irrational 'selfless' behavior can be genetically programmed (hard-coded), so it is not unrealistic that colonial organisms might trade a temporary decrease in growth rate for a long-term gain. For example, cellular slime molds sacrifice themselves during the formation of a colonial fruiting body, even though it is not guaranteed that the spores released by the fruiting body will immediately encounter more favorable conditions. As such, we assess that it is entirely possible that colonial organisms might trade a growth rate of zero for a potentially negative one.

How such 'selfless' behavior could have evolved is beyond the scope of our current study, but is a potential area for future work. One possibility is that nomadic growth rates are not always negative, which could drive evolution to locally optimize towards switching to nomadic behavior when the colonial carrying capacity is reached. Then in harsh times, even when nomadic growth rates are negative, the species would still be able to survive and grow by employing the same switching strategy. As noted in "Discussion", fourth paragraph, and in Appendix Theorem A.7, our proposed switching strategy is beneficial for the species regardless of whether nomadism is modelled as a losing game or not.

Overall, I think the paper is interesting, but needs a much more careful treatment of the biological realism underlying various assumptions, and as a consequence, a much more careful discussion of the biological relevance of the results obtained.

We started with a simpler population model in order to better elucidate the basic mechanisms by which Parrondo’s paradox can occur in species with both nomadic and colonial behaviors. A minimal amount of constraints were imposed to demonstrate the paradox, and so the theoretical results obtained are widely generalizable. As we now show in subsection "Survival and growth under additional constraints", when imposing additional constraints to more accurately model features of certain biological systems, the paradox of survival and long-term growth can still be observed. We believe that we have now satisfactorily responded to the comments made by our reviewers.

*Reviewer #3:*

General Assessment:

The manuscript under review ("Nomadic-colonial alternation can enable population growth despite habitat destruction: An ecological Parrondo's paradox") presents and analyzes a novel model for a population where individuals are one of two behavioral types, which are modelled as subpopulations in an ODE model. Colonial types cooperate, compete and (over the long term) reduce their own carrying capacity. Nomads slowly die off (in the parameter space analyzed). Offspring of each type can "switch" to the other and in the model they do so based on the carrying capacity of the colonial sub-population.Through a careful and detailed analysis, the paper shows that the model demonstrates a paradoxical behavior: in situations where pure colonial or nomadic strategies would die out, switching can result in persistence. Such results are well known in ecology and evolution, but most often occur due to environmental variation [1]. Here, the results occur not because of environmental variation but due to the carrying-capacity-dependent switching rule modeled. This provides a closer analog to Parrondo games than in many ecological studies, which (where it is biologically realistic) might help unify various studies of such reversal behaviors (as called for in [1]).

Thank you reviewer #3 for the close reading. Indeed, many biological studies have drawn a connection to this paradox but often do not incorporate the capital-dependence and history-dependence which are characteristic of Parrondo games. The paradox often emerges due to the presence of exogenous environmental variation. The main contributions of our model are in that it captures capital and history-dependent dynamics within a realistic ecological setting, thereby exhibiting the paradox without the need for exogenous environmental influences.

The particular situation studied in the model of alternation between nomadic and colonial types is of great evolutionary interest as it mirrors both the evolution of cooperative colonies from modular organisms and even the evolution of multicellularity itself. Indeed, the language of ratcheting has been invoked in describing the latter process [2]. More recently, that same group has examined the stabilization of multicellularity through ratcheting in a genetic model ([3] which should be cited and discussed in any revision).

[Libby et al., 2016] ‘Stabilizing multicellularity through ratcheting’ is a highly relevant study and we have included it as a reference to motivate our work. The work is discussed in the Introduction, third paragraph.

The model in the manuscript under review, although not explicitly evolutionary, provides an interesting avenue through which these questions might be explored in the future. However, the behavior demonstrated rests on some peculiarities of the model and its assumptions.

*Concerns:*

Although promising, the model as developed displays some strange and possibly unrealistic behavior, which may be due to atypical assumptions. In particular, modelling the carrying capacity K as a function of the population itself (eqn 3) in a single-species model is not common in ecology (e.g., no such model appears in Table 1 of [4]). One reason may be that, as seen here, K will diverge to infinity when populations become small (e.g., if the colonists die off; Figure 1A).

Thank you reviewer #3 for raising these concerns. Reasoning from first principles suggests that modelling the carrying capacity K as a function of the population is entirely within the realm of possibility, since organisms affect the environment that they live in. Another study that backs up this approach is Yukalov et al., 2009, which demonstrates how this sort of feedback can result in punctuated evolution and growth. Models where K depends on the population size can thus be highly productive for understanding complex ecological phenomena.

Nonetheless, we acknowledge that it is unrealistic for K to diverge towards infinity, assuming a finite habitat size. To address this concern, we have added a new subsection “Survival and Growth under Additional Constraints”, imposing all the constraints suggested by the reviewers. As noted in our response to reviewer #1, saturation of the carrying capacity was implemented by modifying Equation 6 (the reduced version of Equation 4) to incorporate logistic-like behavior. The new dynamics of K are stated in Equation 14. With this equation, the growth of K is limited to a maximal value *K*_max_. K might thus be thought of as the short-term carrying capacity, and *K*_max_ as the long-term carrying capacity.

It is important to note that when *K*_max_ >> K, Equation 14 reduces to Equation 6. This means that all of our previously presented results still hold (and are completely valid), given this condition. Even when *K*_max_ is set to be considerably lower (*K*_max_ = 20), Figures 6 and 7show that survival and long-term growth are still possible. We believe that these results should adequately address the reviewers’ concerns about biological realism.

Vyacheslav I. Yukalov, E. P. Yukalova, and Didier Sornette. Punctuated evolution due to delayed carrying capacity. Physica D: Nonlinear Phenomena, 238(17):1752–1767, 2009

The dynamics of K also play a role in the strange behavior explored in Figures 3 and 4, where a vast overabundance of nomads is produced during a long-lived period of switching, during which K is reduced. Because the switching dynamics (eqn 5) are based on the dynamics of K, and key to driving the analyzed behavior the generality of these behaviors is tied to the reasonableness and generality of this assumption.

After limiting the growth of K to a maximal value *K*_max_, the nomadic growth spikes observed in Figures 3 and 4 can still be observed in Figure 7A. We acknowledge that the presence of growth spikes may be an artifact of the specific model used, and may be unique to biological systems that adhere closely to such a model. However, this does not detract from our broader and more important conclusion that both survival and long-term growth are possible. As our new subsection "Survival and growth under additional constraints" now shows, these remain possible even under slightly different modelling assumptions. In sum, our findings still hold under a full simulation incorporating all of our earlier assumptions and conditions suggested by our reviewers.